# Peritonsillar abscess may not always be a complication of acute tonsillitis: A prospective cohort study

Enni Sanmark[1]*, Johanna Wikstén[1], Hannamari Välimaa[2,3], Leena-Maija Aaltonen[1], Taru Ilmarinen[1], Karin Blomgren[1]

1 Department of Otorhinolaryngology—Head and Neck Surgery, Helsinki University Hospital, Helsinki, Finland, 2 Department of Virology, University of Helsinki, Helsinki, Finland, 3 Department of Oral and Maxillofacial Surgery, University of Helsinki and Helsinki University Hospital, Helsinki, Finland

* enni@sanmark.fi

**Data Availability Statement:** All relevant data are within the manuscript and its Supporting Information files.

## Abstract

The present study aimed to specify diagnostics for peritonsillar abscesses (PTAs) and to clarify the role of minor salivary glands. This prospective cohort study included 112 patients with acute tonsillitis (AT) and PTA recruited at a tertiary hospital emergency department between February and October 2017. All patients completed a questionnaire concerning their current disease. Serum amylase (S-Amyl) and C-reactive protein (S-CRP) levels, tonsillar findings, and pus aspirate samples and throat cultures were analyzed. Eight of 58 PTA patients (13.8%) had no signs of tonsillar infection. The absence of tonsillar erythema and exudate was associated with low S-CRP ($p<0.001$) and older age ($p<0.001$). We also observed an inverse correlation between S-Amyl and S-CRP levels (AT, $r = -0.519$; PTA, $r = -0.353$). Therefore, we observed a group of PTA patients without signs of tonsillar infection who had significantly lower S-CRP levels than other PTA patients. These findings support that PTA may be caused by an etiology other than AT. Variations in the S-Amyl levels and a negative correlation between S-Amyl and S-CRP levels may indicate that minor salivary glands are involved in PTA development.

## Introduction

Acute tonsillitis (AT) is a highly prevalent infection that is responsible for a large number of consultations. Peritonsillar abscess (PTA) is the most common deep head and neck infection, with an incidence of 10–41/100,000 [1–5], and traditionally regarded as a purulent complication of AT, but the evidence for an association between the two is uncertain. PTA may appear after tonsillectomy (TE) without tonsillar remnants, as Windfuhr et al. demonstrated, and the seasonal incidence of the two entities does not follow a symmetrical pattern [1, 6–8]. Over the last three decades, PTA has been speculated to not necessarily arise from AT, but as a consequence of poor dental health, smoking, and salivary dysfunction. Minor salivary glands have been suggested to play a significant role in PTA [3, 8, 9].

**Funding:** This work was supported by the Finnish ORL-HNS Foundation to ES; Foundation of Dorothea Olivia, Karl Walter, and Jarl Walter Perklén's memory (201700022) to ES; and the Helsinki University Hospital Research Fund to KB.

**Competing interests:** The authors have declared that no competing interests exist.

**Abbreviations:** PTA, Peritonsillar abscess; AT, acute tonsillitis; GAS, Group A streptococcus; FN, *Fusobacterium necrophorum*; SAG, *Streptococcus anginosus* group; ID, incision and drainage.

Shared symptoms of AT and PTA are sore throat and fever. PTA patients also suffer from trismus, and the pain is typically asymmetrical [10, 11]. In both AT and PTA, common clinical findings include tonsillar exudate and enlarged cervical lymph nodes. PTA patients also often have asymmetric peritonsillar swelling [11, 12]. The diagnosis of AT and PTA is clinical, but a recent review recommended measuring C-reactive protein (CRP) and performing a full blood count in all PTA patients, though the benefits of laboratory tests in the diagnosis or treatment of PTA are unclear. As dehydration is a common symptom of PTA, electrolytes should also be analyzed [2]. Tachibana et al. reported higher serum leukocyte levels in PTA patients than AT patients, but did not observe differences in CRP levels. In addition, higher serum CRP (S-CRP) levels predicted slower healing after PTA; the mean S-CRP in PTA patients was 85 mg/l (1.3–380 mg/l) [13, 14].

Salivary amylase levels can serve as a marker of salivary function, and recent studies have shown that both serum and pus amylase levels are highly elevated in PTA patients compared to patients with other neck abscesses and dental abscesses. In PTA patients, the mean serum amylase level is 50 U/l and mean pus amylase level 3045 U/l (range 20–11,000 U/l) [15–17].

Group A streptococci (GAS) are the most prevalent bacteria causing both AT and PTA [9–11, 18]. Several studies have reported other bacteria (e.g., other beta-hemolytic streptococci [groups C and G] and *Fusobacterium necrophorum* [FN]) as major pathogens in AT [19, 20], and FN has been recognized as a major pathogen in PTA. In addition, *Streptococcus anginosus* group (SAG) streptococci have been reported as important pathogens in peritonsillar infections and observed to predict the recurrence of PTA [21, 22]. PTAs caused by FN are associated with significantly increased CRP and neutrophil levels compared to PTAs caused by other pathogens [22].

Although the clinical picture of PTAs is diverse, previous studies concerning PTA diagnostics have considered these patients as a uniform group. The aim of this study was to clarify the potential role of minor salivary glands in the development of PTA and to specify diagnostics for PTA.

## Materials and methods

### Participants and samples

The study included 112 patients referred to Helsinki University Hospital Department of Otorhinolaryngology for AT or PTA between February and October 2017. Exclusion criteria were age <15 years or pancreatic disease due to the changes in serum amylase levels caused by pancreatic destruction [23]. Patients were divided into two subgroups based on clinical diagnosis: AT (n = 54) or PTA (n = 58). All patients completed a questionnaire concerning their smoking habits, overall health, current disease, prior antibiotics during the current infection, alcohol consumption, and previous tonsillar or peritonsillar infections. The emergency department physician completed a structured form concerning tonsillar findings, and the physicians evaluated the dental health and oral hygiene (good/poor) by inspecting the oral cavity. Blood samples were collected from all patients, and S-CRP and serum amylase (S-Amyl) levels were analyzed. The PTA patients were also treated with incision and drainage (ID). Pus aspirate samples from PTA patients and throat swabs from AT patients were taken for bacterial culture.

Bacterial culture was performed at Helsinki University Hospital Laboratory Services (HUS-LAB). Pus samples were grown under both aerobic and anaerobic conditions and the superficial throat swabs under aerobic conditions according to the laboratory's standard methods for diagnostic samples. The bacterial culture results were recorded from the hospital laboratory database as reported by the diagnostic laboratory. For the analysis, all isolates of beta-

hemolytic streptococci, SAG and FN, *Haemophilus influenzae*, *Staphylococcus aureus*, and heavy growth of *Neisseria meningitidis* were recorded as separate isolates. Non-FN anaerobic isolates were combined and classified as other anaerobic bacteria. In addition, the reported mixed normal flora was recorded as mixed regional flora.

### Ethical considerations

All procedures that involved human participants were conducted in accordance with the ethical standards of the institutional or national research committee and with the 1964 Declaration of Helsinki and its later amendments or comparable ethical standards. The Ethics Committee of Helsinki University Hospital approved the study protocol. All patients provided written informed consent prior to their participation. According to Finnish legislation, minors over 15 years of age are entitled to give their informed consent without guardian's permission. All provided study information was, however, adjusted for their age group.

### Main outcome measures

Correlations between smoking, alcohol consumption, signs and symptoms of infection, laboratory tests, oral health, and bacterial findings were analyzed and compared between the AT and PTA patients. The main aim of these analyses was to determine whether any differences were present in laboratory tests, oral health, or symptoms between AT and PTA patients. In addition, we examined whether different subgroups of AT or PTA patients exist and whether laboratory values, bacterial findings, certain symptoms, smoking, or oral health are related to a specific subgroup of patients.

### Statistical analysis

Statistical analyses were performed using NCSS 8 statistical software (Hintze, J. [2012]; NCSS 8, NCSS, LLC, Kaysville, UT, USA; www.ncss.com). The FN and FN+SAG groups were combined in the statistical analysis due to the small number of patients in the groups. Numerical variables were analyzed by the Mann-Whitney U-test and Kruskal-Wallis one-way ANOVA. Chi-squared was applied to compare nominal variables. The Spearman rank-correlation was applied to two numerical variables with non-normal distribution. *P*-values $<0.05$ were considered significant.

## Results

A total of 112 patients were included in the study. Patient characteristics and clinical data are presented in Table 1. None of the patients developed a complication during the first 3 months.

### Serum amylase and C-reactive protein levels

A significant inverse correlation was observed between S-Amyl (normal reference 28–100 U/l) and S-CRP (normal reference 0.05–3 mg/l) levels in both the AT and PTA groups (AT, r = -0.519, $p \leq 0.001$; PTA, r = -0.353, $p \leq 0.001$; Fig 1). We found no differences in the S-Amyl or S-CRP levels between the AT and PTA groups (S-Amyl, $p = 0.767$; S-CRP, $p = 0.501$). Alcohol consumption and smoking habits had no effect on S-Amyl levels (alcohol, $p = 0.750$; smoking, $p = 0.205$). No correlation was found between age and CRP (AT and PTA patients r = -0.094, $p = 0.342$; PTA patients r = -0.137, $p = 0.329$). The S-Amyl and S-CRP levels in AT and PTA patients are presented in Table 2.

**Table 1. Patient characteristics.**

| Characteristic | All PTA patients (n = 58) | AT patients (n = 54) | PTA patients without tonsillar findings (n = 8) |
|---|---|---|---|
| Age, years | | | |
| Median | 36 | 28.5 | 50.5 |
| Range | 16–65 | 15–86 | 24–65 |
| Gender | | | |
| Male | 36 (62.1) | 21 (38.9) | 5 (62.5) |
| Female | 22 (37.9) | 33 (61.1) | 3 (37.5) |
| Smoking | | | |
| Non-smoker | 21 (36.8) | 20 (37.0) | 5 (62.5) |
| Smoker | 24 (41.4) | 22 (40.7) | 2 (25.0) |
| Ex-smoker | 12 (20.7) | 12 (22.2) | 1 (12.5) |
| No information | 1 | | |
| Alcohol consumption | | | |
| Yes | | | |
| No | 36 (62.1) | 31 (57.4) | 4 (50.0) |
| No information | 21 (36.2) | 23 (42.6) | 4 (50.0) |
| | 1 | | |
| Prior antibiotics for more than 24 hours | | | |
| Yes | 10 (17.2) | 9 (16.7) | 1 (12.5) |
| No | 48 (82.8) | 45 (83.3) | 7 (87.5) |
| Duration of symptoms [a] prior to PTA | | | |
| 1–3 Days | | | |
| >3 days | | | |
| | 16 (27.6) | 28 (51.9) | 2 (25.0) |
| | 42 (72.4) | 26 (48.1) | 6 (75.0) |
| Unilateral throat pain | | | |
| Yes | 54 (93.1) | 31 (60.8) | 8 (100.0) |
| No | 4 (6.9) | 20 (39.2) | 0 (0) |
| No information | | 3 | |
| Fever | | | |
| Yes | 38 (65.5) | 41 (75.9) | 3 (37.5) |
| No | 20 (34.5) | 13 (24.1) | 5 (62.5) |
| Symptoms of common cold | | | |
| Yes | | | |
| No | 16 (27.6) | 23 (44.2) | 2 (25.0) |
| No information | 42 (72.4) | 29 (55.8) 2 | 6 (75.0) |
| Prior tonsillar infection | | | |
| AT | 13 (22.8) | 12 (22.2) | 0 (0) |
| CT | 1 (1.8) | 1 (1.9) | 0 (0) |
| PTA | 8 (14.0) | 2 (5.6) | 2 (25.0) |
| None | 35 (61.4) | 38 (70.4) | 6 (75.0) |
| No information | 1 | | |
| Tonsillar findings | | | |
| Exudate | | | |
| Erythema | 16 (28.1) | 16 (30.2) | 0 (0) |
| Both | 26 (45.6) | 16 (30.2) | 0 (0) |
| None | 7 (12.3) | 21 (39.6) | 0 (0) |
| No information | 8 (14.0) | 0 (0) | 8 (100.0) |

*(Continued)*

**Table 1.** (Continued)

| Characteristic | All PTA patients (n = 58) | AT patients (n = 54) | PTA patients without tonsillar findings (n = 8) |
|---|---|---|---|
| | 1 | | |
| Oral hygiene | | | |
| Good | | | |
| Poor | 52 (91.2) | 48 (90.6) | 7 (87.5) |
| No information | 5 (8.8) | 5 (9.4) | 1 (12.5) |
| | 1 | 1 | |

Data are presented as n (%) unless otherwise noted. PTA, peritonsillar abscess; AT, acute tonsillitis; CT, chronic tonsillitis

[a] Throat pain, fever, hoarseness.

## Tonsillar findings

In patients with AT, the presence of tonsillar erythema, tonsillar exudate, or both had no effect on S-Amyl ($p = 0.306$) or S-CRP ($p = 0.363$) levels. In the PTA group, eight patients did not have tonsillar erythema or tonsillar exudate. These eight PTA patients had significantly lower S-CRP levels (median 15 mg/l, range 3–40 mg/l) than PTA patients presenting with tonsillar findings (median S-CRP 60 mg/l, range 3–313 mg/l; $p<0.001$). S-Amyl levels were slightly, but not significantly ($p = 0.190$), higher in these eight patients (median 49 U/l, range 27–82 U/l) compared to the other PTA patients (median 41 U/l, range 15–116 U/l; Fig 2). PTA patients without tonsillar findings were significantly older (median age 50.5 years, range 24–65 years) than the other PTA patients (median age 34 years, range 16–60 years; $p = 0.006$). Of the 58 PTAs, two (3.4%) were bilateral. Oral health had no impact on bacterial findings in either group.

a)                                                                                          b)

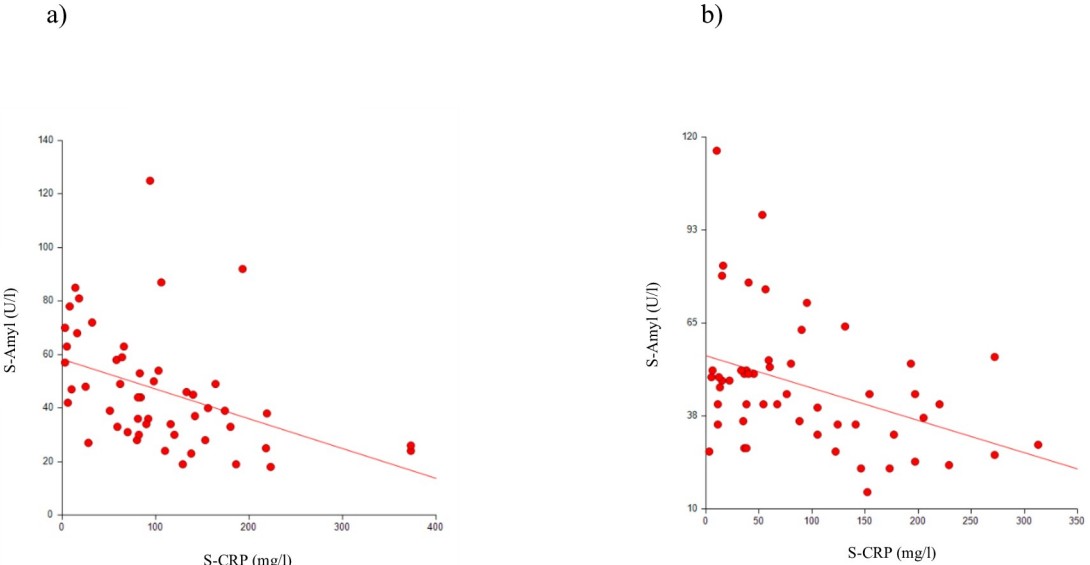

**Fig 1. Correlation between serum amylase and C-reactive protein levels.** (A) Patients with acute tonsillitis (r = -0.519, $p\leq0.001$). (B) Patients with peritonsillar abscess (r = -0.353, p $\leq0.001$).

**Table 2. Serum amylase and CRP levels in AT (n = 52) and PTA (n = 54) patients with different infection symptoms.**

|  | Median amylase in AT (U/l) | Median CRP in AT (mg/l) | Median amylase in PTA (U/l) | Median CRP in PTA (mg/l) |
|---|---|---|---|---|
| Fever |  |  |  |  |
| Yes | 37.5 | 104.5 | 44 | 82 |
| No | 65.5 | 15 | 46 | 40 |
| p-value | 0.0212 | 0.00037 | 0.637 | 0.239 |
| Symptom duration |  |  |  |  |
| <3 days | 39 | 94 | 36 | 74 |
| >3 days | 42 | 80 | 44 | 59 |
| p-value | 0.650 | 0.063 | 0.655 | 0.720 |
| Unilateral pain | 46 | 75.5 | 44 | 60 |
| Bilateral pain | 30.5 | 110 | 32.5 | 120.5 |
| p-value | 0.033 | 0.031 | 0.175 | 0.522 |
| Common cold |  |  |  |  |
| Yes | 41 | 94 | 51 | 36 |
| No | 38.5 | 90 | 41 | 82 |
| p-value | 0.914 | 0.575 | 0.0096 | 0.0755 |
| Oral hygiene |  |  |  |  |
| Normal/Good | 39.5 | 87 | 41 | 67 |
| Poor | 42 | 92 | 48 | 36 |
| p-value | 0.862 | 0.874 | 0.484 | 0.447 |
| History of |  |  |  |  |
| AT | 34 | 90 | 46 | 105 |
| PTA |  |  | 35 | 36 |
| None | 44 | 83 | 44 | 56 |
| p-value | 0.151 | 0.971 | 0.852 | 0.409 |

AT, acute tonsillitis; PTA, peritonsillar abscess

## Bacterial findings

Bacterial samples were taken from all but two PTA patients. The bacterial findings are presented in Table 3. The most common bacteria isolated in AT patients were GAS (41.7% of all samples), group B, C, and G beta-hemolytic streptococci (3.7%), and *H. influenzae* (3.7%). In 51.9% of AT patients, only normal mixed tonsillar flora was reported. In PTA patients, anaerobic Gram-negative rods were a common finding (n = 27, 37.5% of all samples), including FN isolated from five patients. GAS and SAG were isolated from an almost equal number of samples (18.1% vs. 15.3%). In three samples, SAG was found together with FN. *H. influenzae*, *N. meningitidis*, and *S. aureus* isolates were found together with anaerobic Gram-negative rods. *N. meningitidis* growth was heavy and included as a separate finding. One bacterial sample in the PTA group was negative.

The bacteria isolated from the eight PTA patients without tonsillar findings were GAS (n = 2), *S. aureus* (n = 2), FN (n = 1), anaerobic Gram-negative rods (n = 5), and mixed aerobic flora (n = 1). *S. aureus* was isolated together with anaerobic Gram-negative rods in both cases. All of these bacteria were also isolated in other PTA patients; therefore, we found no obvious difference between the two groups in the bacterial etiology of the disease.

No differences were found in the S-Amyl or S-CRP levels between the bacterial findings in either the AT or PTA group (AT, S-Amyl $p$ = 0.620, S-CRP $p$ = 0.331; PTA, S-Amyl $p$ = 0.925, S-CRP $p$ = 0.203). Bacterial findings were not associated with any of the recorded specific

a.

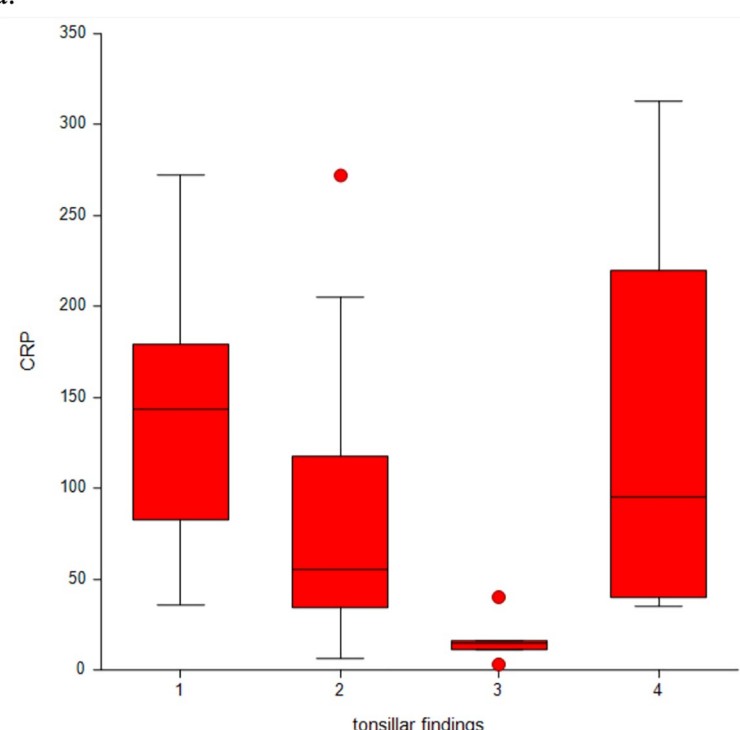

b.

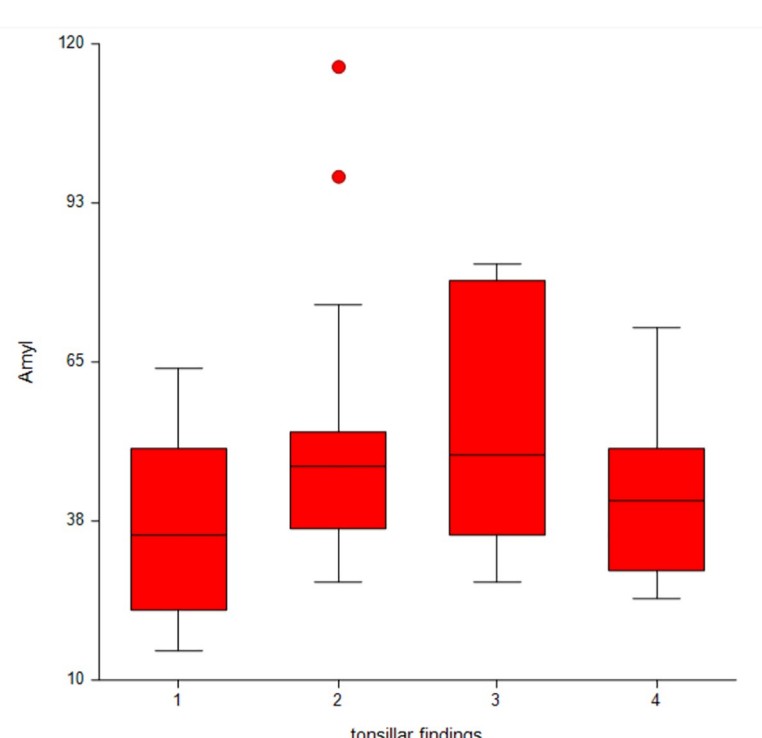

**Fig 2. Distribution of serum proteins in PTA patients with different tonsillar findings.** (A) C-reactive protein (S-CRP) and (B) amylase (S-Amyl). 1, tonsillar erythema; 2, tonsillar exudate; 3, no findings; 4, tonsillar erythema and exudate.

symptoms (unilaterality, absence of common cold, fever) or presence of tonsillar erythema or exudate in the AT or PTA group. Oral health had no impact on bacterial findings in either group.

In patients treated with prior antibiotics, SAG was found more frequently than other bacteria ($p = 0.037$). Patients with SAG were significantly older (median age 45 years, range 27–53 years) than patients with the following other bacteria: GAS (median age 30 years, range 20–58 years), non-SAG/beta-hemolytic streptococci/FN (median age 31.5 years, range 18–65 years), or FN and FN+SAG (median age 25.5 years, range 16–37 years; $p = 0.022$). Seven out of nine patients (77.8%) with SAG were male.

## Discussion

Some of the PTA patients in this study had no tonsillar findings at clinical examination. In contrast, all AT patients had tonsillar erythema, tonsillar exudate, or both. PTA patients presenting without tonsillar erythema and tonsillar exudate had significantly lower S-CRP levels than PTA patients with tonsillar findings. These eight patients without tonsillar findings were also older than other PTA patients. Thus, patient characteristics, S-CRP, S-Amyl, and clinical findings may be tools for differentiating PTA patient subgroups with different etiological factors [3, 5]. In addition, we found an inverse correlation between S-Amyl and S-CRP levels and lower S-CRP and higher S-Amyl levels in afebrile AT patients compared to those with fever.

**Table 3. Bacterial findings in patients with AT or PTA.**

| Variable | | AT[a] | PTA |
|---|---|---|---|
| **Number of samples** | | 54 | 56 |
| **Bacterial findings, n (%)** | | | |
| **Aerobic bacteria** | | | |
| Beta-hemolytic streptococci | | 24 (44.4) | 16 (22.2) |
| | Group A streptococci | 22 (40.7) | 13 (18.1) |
| | Group B, C, or G streptococci | 2 (3.7) | 3 (4.2) |
| *Streptococcus anginosus* group | | 0 (0) | 11 (15.3) |
| *Haemophilus influenzae* | | 2 (3.7) | 1 (1.4) |
| *Neisseria meningitidis* | | 0 (0) | 1 (1.4) |
| *Staphylococcus aureus* | | 0 (0) | 3 (4.2) |
| Other aerobic bacteria | | 0 (0) | 0 (0) |
| **Anaerobic bacteria** | | | |
| *Fusobacterium necrophorum* | | | 5 (6.9) |
| Other anaerobic bacteria[b] | | | 22 (30.6) |
| **Mixed regional flora** | | 28 (51.9) | 13 (18.1) |

AT, acute tonsillitis; PTA, peritonsillar abscess

[a] Only aerobic culture was performed for patients with acute tonsillitis (AT).

[b] Other anaerobic bacteria isolated: anaerobic Gram-negative rods (n = 14), *Prevotella* species (n = 2), *Fusobacterium* species other than necrophorum (n = 4), anaerobic mixed flora (n = 2).

## Strengths and limitations of the study

This prospective study is the first study to compare both S-CRP and S-Amyl between AT and PTA patients, as well as between patients with different signs and symptoms of infection. One limitation of this study is that only aerobic bacteria were analyzed in superficial throat bacterial cultures. In addition, though both aerobes and anaerobes were tested in pus samples, some bacterial findings were so few that it limited the statistical analysis. We did not exclude AT patients with earlier PTA (n = 3) from the study, which could have affected the results, but our cohort included very few of these patients. Patients with alcohol abuse and pancreatic disease were excluded from the study, and none of the patients were pregnant, but patients with other conditions causing elevated S-Amyl, such as gastrointestinal tract infections, eating disorders, drug use, renal dysfunction, and macroamylasemia, were not excluded. Furthermore, the evaluation of tonsillar findings (erythema, exudate) was based on subjective observation by ear, nose, and throat physicians. However, due to prospective nature of the study, as well physicians' substantial experience with tonsillar diseases, the evaluation can be considered reliable. Oral health was also evaluated by an ear, nose, and throat physician, not a dentist, and classified only as good/poor. We used the standard reference values to analyze S-Amyl and did not have the baseline S-Amyl value for the 112 patients. Therefore, it is impossible to determine the actual change in S-Amyl values caused by the infection. Our material is consistent with previous studies concerning patient age, gender, oral health, and smoking habits [2, 9, 19, 24].

## Comparison with other studies

No earlier studies have compared S-CRP and S-Amyl levels between PTA patients with different signs and symptoms. In our study, we observed a group of PTA patients without tonsillar findings. Compared to patients with AT or PTA with tonsillar findings, these patients had lower S-CRP levels and were older. In AT and PTA patients, an inverse correlation between S-Amyl and S-CRP levels was evident. Similar findings have been reported in patients with infection of major salivary glands. Saarinen et al. demonstrated that the majority of patients suffering from parotitis had elevated S-Amyl levels (median 261 U/l, range 24–1220), but only half of the patients had significantly elevated S-CRP levels >40 mg/l; 32% of the patients had normal S-CRP < 5 mg/l (median 13.0 mg/l, range 5–170 mg/l) [25]. These similar findings between our eight PTA patients and patients with confirmed infection of the major salivary glands suggests that the PTA in these eight patients was due to infection of the minor salivary glands [25–28]. One possible reason that S-Amyl levels in PTA patients remain below the reference values is the small size of the minor salivary glands relative to major salivary glands, which prevents the elevation from becoming substantial. In addition, S-Amyl reference values are validated for the diagnosis of pancreatic diseases and may not be directly adaptable to reflect the activity of minor salivary glands. For a more reliable analysis of changes in S-Amyl in PTA patients, we should define the baseline S-Amyl levels in these patients and prefer S-Amyl measurements over salivary amylase.

We found no difference in S-Amyl levels between patients with and without a history of recurrent AT or PTA. This observation differs from an earlier study demonstrating significantly lower pus amylase levels in patients with a recurrent PTA [16]. Tachibana et al. previously showed no difference in S-CRP levels between AT and PTA patients. Our observations were similar [14].

We also observed that afebrile AT patients have lower S-CRP and higher S-Amyl levels than AT patients with fever. One could speculate that salivary gland activation sometimes occurs in AT, though this hypothesis has not been presented previously [4, 29]. AT patients with unilateral symptoms had lower S-CRP and higher S-Amyl levels than patients with bilateral throat

pain, which could also indicate activity in the minor salivary glands. We compared S-Amyl levels between PTA and AT patients and did not observe a difference between the groups. El Saied et al. previously demonstrated a difference in both serum and pus amylase levels between PTA patients and patients with deep neck abscesses and dental abscesses [16, 17].

In our study, GAS was the most prevalent bacterial finding in PTA patients, followed by SAG and FN. In a recent Danish study of PTA, FN was the predominant pathogen, GAS was the second most common, and group C *Streptococcus* was the third [22]. In this study, SAG was found in older patients and predominately in males. Furthermore, SAG has been shown to predict the rapid recurrence of PTA and to be found in older male patients [21, 30].

## Clinical applicability of the study

Our results suggest that the pathogenesis of PTA may differ between patients; thus, patient characteristics, S-CRP and amylase levels, and clinical findings may be useful tools for differentiating these PTA subgroups. For example, when differentiating between subgroups of PTA patients, we could identify patients who would benefit from TE after the first PTA episode. Further studies are needed to clarify the usability of these variables and their significance in helping choose the optimal treatment strategy for different PTA patients.

## Conclusion

The etiology of PTA is not as unambiguous as previously thought. Bacteria may not be the only factor determining the course of the disease. There seems to be a subgroup of PTA patients without signs of tonsillar infection that share features with parotitis. Therefore, it may be possible for PTA in these patients to not begin as a complication of AT, but as an infection of the minor salivary glands. More research is needed to examine the different subgroups of PTA and the role of minor salivary glands in PTA and AT.

## Supporting information

**S1 File.**
(XLSX)

**S2 File.**
(DOCX)

## Acknowledgments

Timo Pessi performed the statistical analysis with the authors.

## Author Contributions

**Conceptualization:** Enni Sanmark, Johanna Wikstén, Karin Blomgren.

**Data curation:** Enni Sanmark, Johanna Wikstén, Karin Blomgren.

**Investigation:** Enni Sanmark, Johanna Wikstén, Karin Blomgren.

**Methodology:** Enni Sanmark, Hannamari Välimaa, Karin Blomgren.

**Resources:** Karin Blomgren.

**Supervision:** Hannamari Välimaa, Karin Blomgren.

**Visualization:** Johanna Wikstén.

**Writing – original draft:** Enni Sanmark.

**Writing – review & editing:** Enni Sanmark, Johanna Wikstén, Hannamari Välimaa, Leena-Maija Aaltonen, Taru Ilmarinen, Karin Blomgren.

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
