## [Decision Letter · Decision Letter 0]

18 Feb 2020

PONE-D-20-00508

Peritonsillar abscess may not always be a complication of acute tonsillitis: A prospective cohort study

PLOS ONE

Dear Mrs. Sanmark,

Thank you for submitting your manuscript to PLOS ONE. After careful consideration, we feel that it has merit but does not fully meet PLOS ONE’s publication criteria as it currently stands. Therefore, we invite you to submit a revised version of the manuscript that addresses the points raised during the review process.

We would appreciate receiving your revised manuscript by Apr 03 2020 11:59PM. To enhance the reproducibility of your results, we recommend that if applicable you deposit your laboratory protocols in protocols.io, where a protocol can be assigned its own identifier (DOI) such that it can be cited independently in the future. For instructions see: http://journals.plos.org/plosone/s/submission-guidelines#loc-laboratory-protocols

We look forward to receiving your revised manuscript.

Kind regards,

Jorge Spratley, MD, PhD

Academic Editor

PLOS ONE

Journal Requirements:

3. Your ethics statement must appear in the Methods section of your manuscript. If your ethics statement is written in any section besides the Methods, please move it to the Methods section and delete it from any other section. Please also ensure that your ethics statement is included in your manuscript, as the ethics section of your online submission will not be published alongside your manuscript.

Reviewers' comments:

Reviewer's Responses to Questions

**Comments to the Author**

1. Is the manuscript technically sound, and do the data support the conclusions?

Reviewer #1: Partly

Reviewer #2: Yes

Reviewer #3: Yes

2. Has the statistical analysis been performed appropriately and rigorously? 

Reviewer #1: Yes

Reviewer #2: Yes

Reviewer #3: Yes

3. Have the authors made all data underlying the findings in their manuscript fully available?

Reviewer #1: Yes

Reviewer #2: Yes

Reviewer #3: Yes

4. Is the manuscript presented in an intelligible fashion and written in standard English?

Reviewer #1: Yes

Reviewer #2: Yes

Reviewer #3: Yes

5. Review Comments to the Author

Reviewer #1: Although a well written paper with a coherent statistical analysis of the data, it is not a new suggestion in the literature regarding the relationship of PTA and the minor salivary glands of Weber (this eponym should have been quoted ...). Other than that, for validation of microbial etiology swabs of the tonsilar surface in the PTA patients should also have been done as for individual comparison of microbial etiology and for comparison with the AT patients. As regarding the s-Amyl levels, the findings could possibly have other causes; in this way, the s-Amyl levels should also have been measured after disease resolution as to exclude false-positive results during PTA crisis.

Reviewer #2: This prospective cohort study of 112 patients nicely compares serum amylase and CRP levels in patients with tonsillitis and peritonsillar abscess, including 8 patients who did not have acute tonsillitis on physical exam. Those who did not have acute tonsillitis but had a PTA had lower CRP and were older. The paper is well-written.

Suggestions:

Would add to keywords; peritonsillar abscess, tonsillitis

Abstract line 11: suggest changing to: "These findings support that some cases of PTA may be caused by an

etiology other than AT. "

Page 11 line 25: Suggest not using the abbreviation TE (eg write out the words)

Page 12 line 4: It is a stretch to say that there is a group of patients who share features with parotitis. Suggest limiting the conclusion to what is known- i.e. the inverse relationship between CRP and S-amyl.

Reviewer #3: interesting topic with good study drawn, intereting number of patines and well designed study

some limitations recognized by the authors and interestingly discussed

good statistical analyses with consistency on results

6. PLOS authors have the option to publish the peer review history of their article (what does this mean?). If published, this will include your full peer review and any attached files.

Reviewer #1: No

Reviewer #2: No

Reviewer #3: Yes: maria helena raposo silveira

---

## [Author Response · Author response to Decision Letter 0]

8 Mar 2020

Reviewer 2: Abstract line 11: suggest changing to: "These findings support that some cases of PTA may be caused by an etiology other than AT. "

Response: Abstract lines 11-12 changed.

Reviewer 2: Would add to keywords; peritonsillar abscess, tonsillitis

Response: peritonsillar abscess, tonsillitis added to keywords

Reviewer 1: Weber´s glands should be quoted 

Response: Weber´s glands are added to introduction, page 4, line 9.

Reviewer 2: Page 11 line 25: Suggest not using the abbreviation TE (eg write out the words)

Response: Page 11 line 4 as suggested

Reviewer 2: It is a stretch to say that there is a group of patients who share features with parotitis. Suggest limiting the conclusion to what is known- i.e. the inverse relationship between CRP and S-amyl.

Response: Page 11, line 15 changed as suggested.

Reviewer 1: Although a well written paper with a coherent statistical analysis of the data, it is not a new suggestion in the literature regarding the relationship of PTA and the minor salivary glands of Weber (this eponym should have been quoted ...). Other than that, for validation of microbial etiology swabs of the tonsillar surface in the PTA patients should also have been done as for individual comparison of microbial etiology and for comparison with the AT patients. As regarding the s-Amyl levels, the findings could possibly have other causes; in this way, the s-Amyl levels should also have been measured after disease resolution as to exclude false-positive results during PTA crisis.

Response: Thank you for your comments. We are not suggesting that the relationship of PTA and minor salivary glands is our original idea. See the introduction Page 4, Lines 6-9: “Over the last three decades, PTA has been speculated to not necessarily arise from AT, but as a consequence of poor dental health, smoking, and salivary dysfunction. Minor salivary glands have been suggested to play a significant role in PTA. “

The idea about validation microbial etiology by comparing the superficial throat swabs between AT and PTA patients is good, and we are definitely considering that in our next study. B

Unfortunately, because we did not take the superficial throat swabs from PTA patients in the present study, the comparison at this stage is not possible .

Other causes that could elevate the S-Amyl levels are listed in discussion: page 9, lines 23-26. The follow up of S-Amyl levels after recovery would increase the reliability of our findings, but because of the nature of prospective study we have no possibility to get this information afterwards. However, by excluding the known causes of elevated S-Amyl levels we can quite reliably assume that the elevated S-Amyl levels are caused by PTA .

---

## [Editor Report · Decision Letter 1]

10 Mar 2020

Peritonsillar abscess may not always be a complication of acute tonsillitis: A prospective cohort study

PONE-D-20-00508R1

Dear Dr. Sanmark,

We are pleased to inform you that your manuscript has been judged scientifically suitable for publication and will be formally accepted for publication once it complies with all outstanding technical requirements.

With kind regards,

Jorge Spratley, MD, PhD

Academic Editor

PLOS ONE
---

## [Editor Report · Acceptance letter]

20 Mar 2020

PONE-D-20-00508R1 

Peritonsillar abscess may not always be a complication of acute tonsillitis: A prospective cohort study 

Dear Dr. Sanmark:

I am pleased to inform you that your manuscript has been deemed suitable for publication in PLOS ONE. Congratulations! Your manuscript is now with our production department. 

With kind regards,

on behalf of

Professor Jorge Spratley 

Academic Editor

PLOS ONE